# Exploring the Accessory Genome of Multidrug-Resistant *Rhodococcus equi* Clone 2287

**DOI:** 10.3390/antibiotics12111631

**Published:** 2023-11-17

**Authors:** Sonsiray Alvarez Narvaez, Susan Sanchez

**Affiliations:** Department of Infectious Diseases, College of Veterinary Medicine, University of Georgia, Athens, GA 30602, USA

**Keywords:** multidrug-resistant, *Rhodococcus equi*, clone 2287, IME2287

## Abstract

Decades of antimicrobial overuse to treat respiratory disease in foals have promoted the emergence and spread of zoonotic multidrug-resistant (MDR) *Rhodococcus equi* worldwide. Three main *R. equi* MDR clonal populations—2287, G2106, and G2017—have been identified so far. However, only clones 2287 and G2016 have been isolated from sick animals, with clone 2287 being the main MDR *R. equi* recovered. The genetic mechanisms that make this MDR clone superior to the others at infecting foals are still unknown. Here, we performed a deep genetic characterization of the accessory genomes of 207 *R. equi* isolates, and we describe IME2287, a novel genetic element in the accessory genome of clone 2287, potentially involved in the maintenance and spread of this MDR population over time. IME2287 is a putative self-replicative integrative mobilizable element (IME) carrying a DNA replication and partitioning operon and genes encoding its excision and integration from the *R. equi* genome via a serine recombinase. Additionally, IME2287 encodes a protein containing a Toll/interleukin-1 receptor (TIR) domain that may inhibit TLR-mediated NF-kB signaling in the host and a toxin–antitoxin (TA) system, whose orthologs have been associated with antibiotic resistance/tolerance, virulence, pathogenicity islands, bacterial persistence, and pathogen trafficking. This new set of genes may explain the success of clone 2287 over the other MDR *R. equi* clones.

## 1. Introduction

*Rhodococcus equi* is an animal and human pathogen mainly known for being the most common cause of severe pneumonia in foals [1,2]. *R. equi* is a soil saprotroph that becomes pathogenic upon acquiring a virulence plasmid (pVAP) that allows *R. equi* to infect and proliferate in macrophages [3]. So far, three different pVAPs have been reported, presenting different host tropisms: equine pVAPA, porcine pVAPB, and ruminant pVAPN [4,5,6]. *R. equi* is endemic on many horse-breeding farms [2,7,8]. It mainly produces disease in foals between 1 and 4 months of age, typically in the form of a multifocal purulent pneumonia [9,10]. The pulmonary infection is contracted during the neonatal period via inhalation of contaminated aerosolized dust [11,12,13]. No commercial vaccines or other consistently efficient immunoprophylactic strategies are available [14,15]. The elevated costs resulting from veterinary care, long-term therapy, and mortality of foals on endemic farms have forced many breeders to adopt practices such as chemoprophylaxis for all foals during the first weeks following birth [16] or to treat foals presenting ultrasonographic lung lesions with antimicrobials before the onset of clinical signs [17,18]. Only a few antibiotics have been seen to be clinically effective in horses to treat *R. equi* [19,20], and the combination of a macrolide and rifampin has been the treatment of choice since the 1980s [21,22,23]. Before the early 2000s, *R. equi* resistant to macrolides and rifampin were rarely found [24]. Then, the practice of thoracic screening plus subclinical treatment was implemented, resulting in a considerable increase in the use of macrolides and rifampin on endemic farms and the subsequent emergence of MDR *R. equi* isolates resistant to these two antibiotics [25].

Over time, the constant antimicrobial selective pressure exerted by the continuous prophylactic antimicrobial treatment has promoted the spread of multidrug-resistant (MDR) *R. equi* [24,25,26]. To date, isolates resistant to macrolides and/or rifampin have been reported in 16 US states [24], China [6,27], Ireland [28,29], France [30], and Poland [31]. Our previous work revealed that in the US, *R. equi* isolates resistant to macrolides and rifampin are mainly clustered in three clonal populations: clone 2287, clone G2016, and clone G2017 [26,32,33]. Clone 2287 harbors pRErm46, a 90 kb conjugative plasmid that carries antimicrobial resistance genes (ARGs) for macrolides, lincosamide, streptogramin B (MLS_B_), tetracycline, and sulfamethoxazole [34] and a chromosomal *rpoB^S531F^* mutation conferring resistance to rifampin. pRErm46 horizontal gene transfer (HGT) into a different *R. equi* genetic background gave rise to clone G2016, associated with a different rifampin resistance mutation (*rpoB^S531Y^*). These two clones are the only resistant ones recovered so far, despite an intense search of sick horses, with clone 2287 accounting for most of the clinical cases [26]. Clone G2017, carrying MLS_B_ resistance plasmids pRErm51 and pMobErm51, has been found exclusively in the environment [26,33]. Although the presence of pVAPA (a conjugative plasmid that carries the virulence factors required to colonize the equine host efficiently) has been reported in members of the three MDR clonal populations, clone 2287 is the most-recovered MDR isolate from sick animals [35,36]. Here, we investigated the accessory genome of 207 *R. equi* (environmental and clinical isolates) to elucidate the genetic mechanisms that make clone 2287 superior at infecting foals.

## 2. Materials and Methods

### 2.1. Bioinformatic Analysis

BLAST+ v2.9.0 [37] was used to align the contigs of 207 *R. equi* isolates characterized in our earlier studies [32,33,35] to known *R. equi* genetic elements (*R. equi* chromosome, pVAPA, pRErm46, and *erm*(51)) at >95% identity and >80% coverage. Similarly, BLASTn [37] was used to align the unknown contigs from sample 156 to the other novel contigs and the nucleotide NCBI database. IME2287 annotation was performed using Prokka V1.14.5 [38] and InterProScan V86.0 [39] and subsequently manually inspected and curated based on BLASTx [37] analysis.

### 2.2. Bacterial Strains and Culture Conditions

Two *R. equi* strains were used in this study: *R. equi* PAM2287 (NCBI BioSample database no. SAMN04880532), which is a macrolide- and rifampin-resistant clinical isolate carrying virulence plasmid pVAPA, macrolide resistance plasmid pRErm46, rifampin mutation *rpoB^S531^*, and mobilizable element IME2287 [35], and *R. equi* 103^-^Apra^R^, which is a plasmidless derivative strain of reference strain *R. equi* 103 containing the *aac*(3)IV apramycin resistance cassette integrated on the chromosome [40]. *R. equi* isolates were routinely cultured in brain heart infusion medium (BHI; Difco Laboratories-BD, Franklin Lakes, NJ, USA) at 30 °C and 200 rpm unless otherwise stated. Agar media were prepared by adding 1.6% of bacteriological agar (Oxoid, Basingstoke, United Kingdom). Media were supplemented with antibiotics (erythromycin, 8 μg/mL; apramycin, 50 μg/mL; rifampin, 25 μg/mL; Sigma, Saint Louis, MO, USA) when required. All in vitro bacterial work (including the bacterial conjugation and the plasmid loss assays) were carried out in our laboratory at the Athens Veterinary Diagnostic Laboratory from the University of Georgia (Athens, GA, USA).

### 2.3. Bacterial Conjugation

Conjugation assays were carried out as described previously by Alvarez-Narvaez et al. [41]. Briefly, *R. equi* donor (MDR *R. equi* clone 2287) and recipient (macrolide-susceptible 103^-^Apra^R^) strains were grown overnight in BHI and in the presence of the corresponding antibiotic. Then, donor and recipient bacteria were mixed 1:1 in 5 μL of BHI and spotted in a thick drop onto a BHI plate. After 72 h of incubation at 30 °C, the bacterial mixture was scraped and resuspended in PBS. Serial dilutions were plated onto BHI agar supplemented with apramycin (recipient selection) or apramycin plus erythromycin (transconjugant selection). Conjugation ratios were calculated using the following formula: conjugation ratio = no. of transconjugant cells/no of recipient cells.

### 2.4. Plasmid Loss Assay

*R. equi* PAM2287 was inoculated into 10 mL of BHI, or donor horse serum (DHS), and grown at room temperature (RT, ~22 °C) or 37 °C with shaking (200 rpm) for 24 h. After that, bacteria were subcultured in 10 mL of fresh BHI or DHS at an initial optical density at 600 nm wavelength (OD_600_) of 0.02. OD_600_ was measured using a Thermo Scientific™ Multiskan™ FC Microplate Photometer (Thermo Fisher Scientific, Waltham, MA, USA). Every subculture was considered a passage, and 45 passages were performed in this experiment. At passages 0, 15, 30, and 45, cultures were serially diluted 1:10 in PBS and plated onto BHI plates. In parallel, *R. equi* PAM2287 was inoculated into 5 mL test tubes (*n* = 10) containing 1 g of autoclaved soil (121 °C for 20 min; Ref. [42]) from the gardens of the University of Georgia School of Veterinary Medicine at a concentration of 5 × 10^4^ CFU/g. Inoculated tubes were mixed well (shaking at ~1800 rpm in a vortex genie 2 bench mixer (Scientific Industries, Bohemia, NY, USA) for 30 min) and incubated statically at RT (*n* = 5) and 37 °C (*n* = 5) for up to 45 days. On days 0, 15, 30, and 45, the soil of 1 tube per temperature condition was resuspended in 2 mL phosphate-buffered saline (PBS) (by shaking at ~1800 rpm in a vortex genie 2 bench mixer (Scientific Industries) for 30 min) and quantitatively cultured through serial 10-fold dilutions in BHI. One hundred colonies per medium (BHI, DHS, and soil) and temperature (RT and 37 °C) were randomly selected and re-plated onto BHI, BHI-Ery, and BHI-Rmp plates. The presence of pRErm46 IME2287 and pVAPA was tested using PCR.

### 2.5. PCR

Transconjugants were confirmed using PCR for the macrolide-resistant gene *erm*(46) and apramycin-resistant gene *aac*(3)IV. pVAPA, pRErm46, and IME2287 mobilization were also tested using PCR. Table 1 contains the list of oligos used in this study. PCRs were carried out using a T100 thermocycler (Bio-Rad, Hercules, CA, USA) and GoTaq Flexi DNA polymerase (Promega, Madison, WI, USA) under the conditions specified in Alvarez-Narvaez et al. [41]. Briefly, we used an initial denaturation step for 5 min at 95 °C, followed by 30 amplification cycles (involving 30 s at 95 °C of denaturation, 30 s of oligonucleotide hybridization at the appropriate melting temperature, and 2 min of elongation at 72 °C) and a final elongation of 10 min at 72 °C. 

## 3. Results

### 3.1. The Search for Genetic Elements in MDR R. equi Clone 2287 Selected and Maintained over Time

PacBio and Illumina whole-genome assemblies from 62 *R. equi* clinical isolates (40 MDR clone 2287 and 22 susceptible) collected between 2002 and 2017 in different US states and characterized in our earlier study [32] were used to identify potential new genetic elements associated exclusively with MDR *R. equi* clone 2287 (Appendix A). Four thousand and one contigs from the 40 members of clonal population 2287 were first aligned to known *R. equi* genetic elements (chromosome, virulence plasmid pVAPA, and MDR plasmid pRErm46). In total, 2669, 85, and 168 contigs matched with the *R. equi* chromosome (accession number NC_014659), pVAPA1037 (accession number NC_011151), and pRErm46 (accession number KY494640), respectively, with >95% similarity and >80% coverage, and 1079 contigs were identified as unknown.

Strain 156 showed fewer unknown contigs, only two: contig_221 and contig_249. These two contigs were aligned to all the unknown contigs from the other resistant strains and to the genomes of the 18 susceptible isolates. Only contig_249 was common in all resistant strains and was not present in the assemblies of any susceptible isolates. Contig_249 was 22,578 bp long and comprised a novel 14,317 bp genetic element (Figure 1) that we noted, IME2287 (integrative mobilizable element from clone 2287), and a subsequent partial IME2287 duplication of 8260 bp in length. Further analysis of the contigs carrying this novel piece of DNA showed that in most of the cases, IME2287 was the only genetic element that appeared in the contigs, repeated (and sometimes inverted) in tandem (Figure 2). Further analysis showed that only one isolate (sample 169) presented with IME2287 randomly inserted in the bacterial chromosome (Figure 2).

### 3.2. Molecular Characterization of IME2287, a Putative Mobilizable Element Associated with Macrolide Resistance in MDR R. equi

The in silico annotation of IME2287 identified 20 ORFs (Figure 1), of which 11 were predicted to have a biological function or a functional domain and 9 were classified as hypothetical proteins. Homology searches showed that the novel set of genes carried by IME2287 share high homology with genes previously found in the genus *Rhodococcus* and other *Actinobacteria*. Table 2 lists the 20 ORFs with the in silico functional predictions and corresponding homologies. Of interest, four genes are clustered together and code for proteins involved in DNA replication and partitioning. In this order, ORF IME2287_0010 was predicted to be a ParG-like DNA binding protein with a ribbon–helix–helix domain in its C-terminal region, ORF IME2287_0020 was identified as an AAA-ATPase ParA partitioning protein, ORF IME2287_0030 was classified as a RepB-like RNA polymerase with a DNA binding domain, and ORF IME2287_0050 was found to be a RepA plasmid replicase. These genes indicate that IME2287 is most likely a self-replicative genetic element.

ORF IME2287_0170 encodes a serine recombinase that would allow IME2287 to excise and reintegrate into the *R. equi* genome. A maximum identity value of 87% with its closest homolog unclassified ISBli29 from *Brevibacterium linens* (*Actinobacteria*) indicates that the serine recombinase is a novel orphan transposase, which we named ISRe2287. The presence of this gene and the replication and partitioning machinery suggested that IME2287 could be an integrative conjugative element (ICE). However, no conjugation genes were predicted in IME2287, and we found the inverted repeat (IR) sequence 5′-CAATCATTCCTTACAGCAAGTCAGCTTGTT-3′ invariably followed by the directly repeated (DR) tetranucleotide TTAC, supporting that IME2287 could be a transposable element. Additionally, IME2287 carries a gene that encodes a protein containing a Toll/interleukin-1 receptor (TIR) domain (TcpR, IME2287_0140) that acts by suppressing the innate immune system in other bacteria species [45,46,47,48] and a putative toxin–antitoxin system composed of a PIN domain-containing protein (toxin, IME2287_0150) and a MerR-like DNA binding protein (antitoxin, IME2287_0160).

### 3.3. IME2287 Moved from MDR R. equi Clone 2287 to other R. equi Genetic Backgrounds Associated with Macrolide Resistance

To explore if IME2287 was a mobile genetic element, the whole-genome assemblies of 10 additional MDR *R. equi* clinical isolates harboring pRErm46 but with genetic backgrounds different from clone 2287 were analyzed. Two isolates were members of the new MDR *R. equi* clonal population G2016 (Table 3), and eight were classified as singletons (an MDR *R. equi* isolate that shows a unique genetic background and therefore does not belong to any of the known *R. equi* clonal populations) in previous phylogenetic analysis [26,32]. We observed the presence of IME2287 in one of the *R. equi* G2016 clones and two of the singletons, indicating that this element moved from *R. equi* clone 2287 into other *R. equi* isolates through horizontal gene transfer (HGT).

Furthermore, we looked for evidence of IME2287 in the PacBio whole-genome assemblies from 135 *R. equi* environmental isolates collected from 100 farms in central Kentucky for a previous investigation [26,33]. The 45 environmental MDR *R. equi* that previous phylogenetic analysis [26,33] classified as part of clonal population 2287 were shown to harbor both pRErm46 and IME2287 in their genome, while none of the 38 susceptible isolates did (Table 3). *R. equi* clone G2016 (sample 52), the only MDR environmental isolate of this clone, carried pRErm46 but was not shown to have IME2287. Similarly, only 7 of the 39 MDR *R. equi* environmental isolates phylogenetically classified as members of the clonal population G2017 were shown to carry pRErm46/tnRErm46. Of these seven, three also carried IME2287, and four did not. Regarding the environmental singleton-resistant isolates, nine out of twelve were seen to carry pRErm46 or tnRErm46, and from those nine, seven isolates also carried IME2287 (Table 3).

As previously seen in the clinical isolates, IME2287 always appears in an individual contig that exclusively contains copies of these elements (Figure 2). Interestingly, we did not find any clinical or environmental isolates that presented IME2287 without any genetic elements (pRErm46 or tnRErm46) associated with the *erm*(46) resistance gene, suggesting that IME2287 could be hijacking pRErm46 conjugation machinery to move.

### 3.4. IME2287 Mobilization Is Independent of Macrolide Resistance Plasmid pRErm46

We previously reported that macrolide resistance plasmid pRErm46 was mobilizable at a high frequency through horizontal gene transfer via bacterial conjugation [35,41]. The findings above lead to the following question: is IME2287 mobilizable simultaneously with pRErm46? To answer this question, conjugation assays were carried out using MDR *R. equi* clone 2287 as a donor and susceptible (pRERrm46- negative) avirulent *R. equi* 103^-^ with an apramycin resistance *aac*(3)IV cassette (103^-^Apra^R^) [40] as the recipient in a ratio 1:1. Transfer of pRErm46 to 103^-^Apra^R^ was observed at a frequency of 2.06 ± 0.70 × 10^−3^ transconjugants/recipient cells. In total, 90 CFUs were PCR-tested for virulence plasmid pVAPA and novel element IME2287. Five out of the ninety colonies tested were shown to have acquired pVAPA together with pRErm46, but no colonies were carrying IME2287, indicating that its mobilization is most likely independent from pRErm46 conjugal transfer.

Next, we investigated if IME2287 is lost over time in the same fashion pRErm46 is. We monitored changes in the antimicrobial resistance phenotype and genotype of MDR *R. equi* clone 2287 culture-passed daily in BHI and DHS at RT (~22 °C) or 37 °C for 45 passages. In parallel, we looked for changes in the antimicrobial resistance phenotype and genotype of MDR *R. equi* clone 2287 incubated in autoclaved soil at RT or 37 °C for 45 passages (Table 4). We observed that the percentage of macrolide- and rifampin-resistant colonies of clone 2287 was maintained for the entire experiment duration (45 passages) in all conditions, except for bacteria incubated in DHS at 37 °C, where macrolide resistance declined progressively over time. This indicates that, regardless of the temperature in BHI and soil, macrolide and rifampin resistance expression do not represent a fitness cost for *R. equi* clone 2287.

We PCR-tested the 46 CFUs that were shown to have lost macrolide resistance for the presence of pRErm46 and IME2287 (Table 5). As expected, all susceptible colonies had lost the macrolide resistance plasmid, but only half of those (*n* = 23) also lost IME2287, indicating again that the mobilization of these two elements is independent. Interestingly, looking at IME2287 loss over time, we realized that up to passage 30, the colonies that lost pRErm46 still maintained IME2287. In passage 45, 23 out of 36 (~64%) appeared to have lost both elements. This indicates that the loss of IME2287 is most likely subsequent to the loss of the pRErm46 plasmid.

## 4. Discussion

The emergence and spread of MDR bacterial clones represent a significant threat to animal and human health. The success of these dominant bacterial clades relies on the acquisition of virulence and antimicrobial resistance (AMR) genes associated with mobile genetic elements [49]. Although we currently know more about resistance mechanisms and their mobilization dynamics, there is still much to learn to help us prevent their selection and spread. Here, we explored the accessory genome of MDR *R. equi* clone 2287, looking for genetic elements that could explain its superiority over other MDR R. equi clones. Considering that bacterial genomes are subject to genetic drift that deletes superfluous sequences [50], we hypothesized that any genetic element that would provide an advantage in the adaptation of MDR clone 2287 would be exclusively present in all clone members and maintained over time. Based on this, we first performed a preliminary screening of clone 2287’s accessory genome, trying to identify genetic elements present in all the genomes of the clonal population and not in susceptible strains. We only identified one genomic sequence under this threshold, and its genomic annotation indicated that we were dealing with a putative mobile genetic element. Consequently, we named the novel element IME2287. 

The oldest member of clone 2287, in which we detected the presence of IME2287, is an isolate collected in Florida in 2002. Since then, IME2287 has been found in all the *R. equi* phylogenetically classified as clone 2287 that were analyzed (*n* = 85, over a period of 15 years in five different US states); the most recent isolate was found in Kentucky in 2017. It is not surprising to find the *R. equi* genome harboring new genetic elements because this bacteria species is a unique example of plasmid-driven full virulence, antimicrobial resistance, and host adaptation [10]. However, *R. equi* tends to lose its virulence host-adapted plasmids (pVAPs) when living as a saprotroph in the soil as they introduce a fitness cost due to the expression of the virulence genes [5,51]. The fact that IME2287 has been maintained for more than 15 years in this clonal population is intriguing and suggests that instead, IME2287 may not impact *R. equi* fitness, or if it does, the role that IME2287 plays for the bacterium overcomes that fitness cost.

We performed a deep genetic characterization of IME2287, looking for genes that could explain its persistence over time. We found that IME2287 carries a potential new virulence factor. The *tcpR* gene encodes a putative TIR domain protein that we designated as TcpR (Figure 1, Table 2). In eukaryotes, TIR domains are part of Toll-like receptors (TLRs), a family of proteins that recognize pathogens and initiate the innate immune response [52]. TIR domain proteins are also found in plants, where they mediate disease resistance, and in bacteria, associated with virulence and bacterial metabolic regulation [47]. TIR domain proteins have been reported to block TLR signaling, inhibiting innate immune responses in important animal and human pathogens such as *Brucella* spp. [53], uropathogenic *Escherichia coli* (UPEC) [54], nosocomial strains of *Enterococcus faecium* [55], and *Staphylococcus aureus* [48]. TIR domain proteins are frequently carried on mobile genetic elements and, in some cases, are associated with ARGs [48]. The innate immune system is the principal response against intracellular *R. equi* [56]. Therefore, an extra virulence factor, such as TcpR, would help *R. equi* circumvent innate host immunity during infection, which is extremely important for the bacteria. It would explain why IME2287 is still within all MDR *R. equi* clinical isolates 15 years after being first reported. Furthermore, this extra virulence factor could be the key to clone 2287’s superiority at infecting animals over the other genomic backgrounds.

Another interesting operon also found in IME2287 consists of two genes that encode an MerR transcriptional regulator and a PIN domain ribonuclease (MerR-PIN). In silico work has classified MerR-PIN operons as putative Type 2 toxin–antitoxin (TA) systems [57], in which the PIN ribonuclease is toxic to the cell and stable. At the same time, the MerR antitoxin is unstable and requires continuous transcription to inhibit the produced toxin [58]. MerR-PIN and other Type II TA systems are often found in plasmids or, as appreciated in IME2287, inserted in chromosomes in association with mobile genetic elements [57,59]. TA systems have been associated with antibiotic resistance, virulence factors, and pathogenicity islands in pathogenic bacteria [58]. One of the first functions assigned to TA systems was plasmid stabilization through a process known as “post-segregational killing”, a suicide mechanism for those cells that do not carry the TA plasmid after cell division [57]. More recently, Type II TA systems have been described as involved in bacterial permanence under stress conditions such as nutrient starvation or antibiotic exposure [60]. Under stress conditions, the antitoxins are selectively degraded, leaving toxins to promote growth arrest and dormancy. Once the stressor is removed, dormant persistent cells revert to the actively growing state and repopulate the original population [60]. Based on the current information regarding Type II TA systems, we hypothesized that the MerR-PIN system could have a double function for MDR *R. equi,* (i) avoiding the loss of IME2287 during cell division for over 15 years and (ii) mediating the permanence of MDR *R. equi* in the soil even in the absence of antimicrobial selective pressure. Future in vitro experimentation to determine MerR-PIN functionality is guaranteed.

IME2287 has been observed in several different genetic backgrounds, indicating that this genetic element can move through horizontal gene transfer. The IME2287 genetic characterization revealed that IME2287 carries a replication and partitioning operon (Figure 1, Table 2) and an integration/excision capability via a serine recombinase. This is consistent with the core modules of integrative conjugative elements (ICEs) previously identified in Actinomycetes [61]. However, IME2287 lacks the conjugation module, often consisting of a relaxase, a coupling protein, and a mating pair formation system [61]. We also identified IR and DR flanking IME2287, suggesting that this genetic element is likely a mobilizable transposon or a non-canonical IME [62]. The distinction between mobilizable transposons and IMEs is unclear and is considered irrelevant for some [62]. Various non-canonical IMEs (that do not encode any relaxase) have been identified in the Firmicutes and Proteobacteria phyla, sometimes with several copies integrated in tandem, like in the case of IME2287 [63]. Regardless of the nature of IME2287 (transposon, or IME), the in silico data obtained in this project reflect that at some point, IME2287 jumped from clone 2287, its native genomic background, to other *R. equi*. IME2287 was found in 50% (13/27) of pRErm46 containing non-clone 2287 *R. equi,* suggesting that its mobilization could be tied to the antimicrobial resistance plasmid mobilization. Hence, we hypothesized that IME2287 (lacking its conjugal machinery) could hijack the pRErm46 conjugation apparatus to spread. We explored this possibility using mating experiments, and we found that none of the transconjugants that acquired pRErm46 also received IME2287. This indicates that IME2287 mobilization is likely independent of pRErm46. However, we cannot discard that IME2287 could indeed be using pRErm46 mobilization machinery but with a very low mobilization rate, or maybe IME2287 transfer is not triggered or selected by exposure to macrolides.

Given the lack of success of our IME2287 transfer attempt, we decided to test if the loss of this novel genetic element was tied to the loss of pRErm46. Our experiments subculturing clone 2287 in the absence of antimicrobials and under different media and temperature conditions showed that the loss of IME2287 is independent of pRErm46 and that the recently discovered genetic element would get lost, most probably after the loss of the macrolide-resistant plasmid. Additionally, this last experiment allowed us to study the persistence of clone 2287 resistant phenotypes without selective pressure. We observed that clone 2287 maintained a rifampin resistance phenotype during the entire duration of the experiment, independent of the incubation conditions. Willingham-Lane and collaborators studied the stability of three lab-generated rifampin-resistant *rpoB* mutations in the *R. equi* 103S genetic background by passing bacteria in BHI at 37 °C [64]. They reported that two of the three *rpoB* mutations reverted to the wild-type form after 20 passages (>90% revert) and that these mutations resulted in decreased growth in vitro and in soil. This, together with the fact that the mutation present in MDR clone 2287 (*rpoB^S531F^*) has been impossible to recreate in the laboratory (Willingham-Lane personal communication), suggests that *rpoB^S531F^* has been naturally selected over other potential *rpoB* mutations because it does not impact *R. equi* fitness either in vitro (soil and BHI at RT) or in vivo (DHS and 37 °C). In our experiment, clone 2287 maintained its macrolide resistance phenotype when incubated at room temperature independently of the medium. We only found a significant reduction (of ~35%) in macrolide resistance when clone 2287 was subcultured in horse serum at 37 °C. Willingham-Lane et al. also investigated the stability of macrolide resistance in *R. equi* 103S subcultured in BHI at 37 °C and the performance of the macrolide-resistant strain in vitro and soil [64]. Under their conditions, macrolide resistance was lost over time (decrease ~50% after 40 passages), and the growth of the macrolide-resistant strains was affected in vitro and in soil. We did not observe a decrease in the macrolide-resistant phenotype in BHI over time, but the duration of our experiment was shorter. Additionally, our previous work did not detect a fitness cost associated with the presence of pRErm46 in *R. equi* 103S [35]. We decided to subculture bacteria in DHS and soil at different temperatures to recreate the closest conditions *R. equi* will face inside the host (DHS at 37 °C) and in the environment (soil at RT). Taking this into consideration, our results indicate that in the environment (i.e., bacteria in soil), the presence of pRErm46 does not have an impact on the growth of clone 2287, which will be able to keep the macrolide-resistant plasmid for long periods in the absence of antimicrobial selective pressure. However, inside the host, when antimicrobials are not administered (DHS at 37 °C), the expression of the virulence genes is essential for *R. equi* survival at the expense of reduced growth [3], which will prompt the bacteria to lose the non-essential MGEs such as pRErm46 and IME2287.

## 5. Conclusions

The work presented herein studied a clonal competitive event in the zoonotic human pathogen *R. equi*. We characterized an accessory genetic element, IME2287, in the genome of MDR *R. equi* clone 2287 that could explain its superiority over other MDR *R. equi* clones. More specifically, this study elucidated at least two molecular traits in IME2287, a toxin–antitoxin (TA) system and a host NF-kB signaling inhibitor, that successful MDR *R. equi* clones may acquire to their advantage when infecting and colonizing the host. Unfortunately, the factors that trigger IME2287 mobilization and the genetic mechanisms required for this task are still unknown.

## Figures and Tables

**Figure 1 antibiotics-12-01631-f001:**
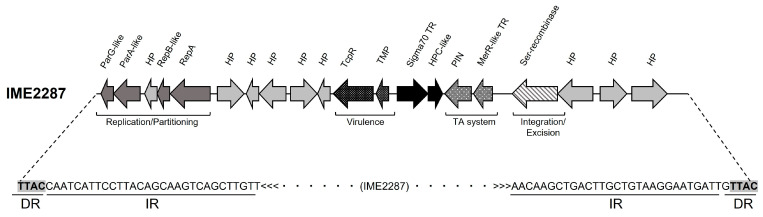
Genetic structure of IME2287. IME2287 carries its own replication and partitioning machinery (dark grey) comprised of ParG-like, ParA-like, RepB-like, and a RepA proteins in the same operon and excision/integration capacities via a serine recombinase (white with grey lines). Other exciting genes in IME2287 include the following: a putative virulence operon (black with white polka dots) composed of a Toll/interleukin-1 receptor protein (TcpR) and a transmembrane protein (TMP) and a putative toxin–antitoxin system containing a PIN domain-containing protein (toxin) and a MerR-like DNA binding protein (antitoxin). See text and Table 2 for other IME2287 components. DRs, direct repeats, (shaded) at the junction with genomic DNA and adjacent inverted repeats (IRs) comprise the TTAC sequence targeted by IME2287. This figure was produced using SnapGene Viewer v7.0.3 (www.snapgene.com) and manually edited with Microsoft PowerPoint for Mac v16.78.3.

**Figure 2 antibiotics-12-01631-f002:**
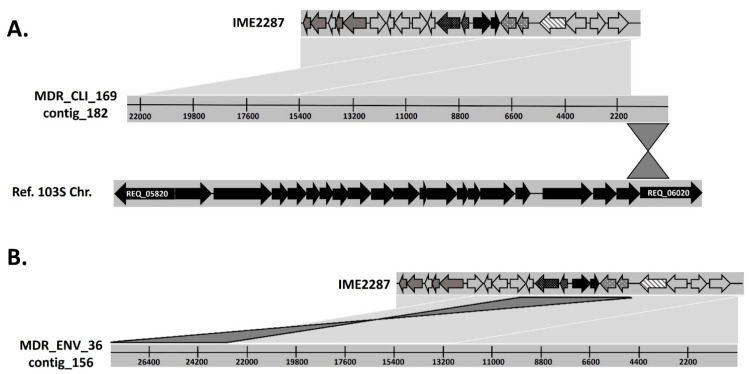
(**A**) IME2287 insertion in *R. equi* chromosome. Above, IME2287 with ORFs are represented by arrows. In the middle, contig 182 from MDR clinical isolate 169 (GenBank accession no. SAMN13392202). Below, a portion of the *Rhodococcus equi* 103S chromosome (ORFs are represented by black arrows; GenBank accession no. FN563149.1). (**B**) Example of IME2287 duplications. Above, IME2287 with ORFs represented by arrows. Below, the environmental MDR isolate 36 (GenBank accession no. WVCU00000000). Regions with significant similarity (nBlast, Score matrix Blosum62) are connected by colored lines (light grey, sequences in direct orientation; dark grey, sequences in reverse orientation). All connecting lines between sequences represent a percentage of identity > 97%. This figure was produced using the Artemis Comparison Tool (ACT) [44] and manually edited with Microsoft PowerPoint.

**Table 1 antibiotics-12-01631-t001:** List of oligos used in this study.

Name	Sequence 5′–3′	Amplicon Size (bp)	Purpose	Source
IME2287_repAB_F	GGAGCACTACTACTGGACG	1746	IME2287 backbone marker	[43]
IME2287_repAB_R	GTTGACTGTGAACTCGGTGT			
IME2287_sigma70_F	CTTGCGAGTAGGACATGAAG	1752	IME2287 backbone marker	[43]
IME2287_sigma70_R	GACCTTCGTCAGGGAGTAAG			
IME2287_tnpR- helix_F	TCTACGTCGACAAGAAGTCC	1745	IME2287 backbone marker	[43]
IME2287_tnpR- helix_R	GTATGTGAACCGACCTTGTG			
ChoE_F	AGTTGTCGATTCCCATCGTC	672	*choE* gene, chromosomal marker	[5]
ChoE_R	AAGCGCAACTACTTCGAGGAG			
TraA-F1	AGAGTTCATGCGTGACAACG	959	*traA* gene, pVAPA backbone marker	[5]
TraA-R1	GTCCACAGGTCACCGTTCTT			
*erm*(46)F	TATGGAGTCGATCTGCAACG	1098	macrolide resistance gene *erm*(46)	[34]
*erm*(46)R	GAGATCGGACGAGTCTGACA			
pRErm46_traG_F	ACCGTCGTAGCAGTAGCC	1533	*traG* gene, pRErm46 backbone marker	[35]
pRErm46_traG_R	CCTCAGCGAGTGTCTTCTC			
ApraF	GGCCACTTGGACTGATCGAG	937	apramycin cassette *aac*(3)IV inserted in chromosome	[40]
ApraR	GCATGACCGACTGGACCTTC			

**Table 2 antibiotics-12-01631-t002:** Annotation of IME2287 from *R. equi* sample 156.

IME2287 Locus Tag	Location (nt Position)		Size (bp)	Product (BlastX)	Coverage (AA Level)	Identity (AA Level)	Species
direct_repeat	1	4	3	-	-	-	-
inverted_repeat	5	34	29	-	-	-	-
IME2287_0010	46	303	257	ParG	>90%	>80%	Conserved in *Actinobacteria*
IME2287_0020	306	1067	761	ParA	>50%	>50%	Conserved in *Actinobacteria*
IME2287_0030	1165	1488	323	Conserved hypothetical protein	>90%	>80%	Conserved in *Rhodococcus* spp.
IME2287_0040	1478	1765	287	RepB	>70%	>80%	Conserved in *Rhodococcus* spp.
IME2287_0050	1765	2651	886	RepA	>70%	>70%	Conserved in *Rhodococcus* spp.
IME2287_0060	2873	3520	647	Conserved hypothetical protein	>70%	>50%	Conserved in *Actinobacteria*
IME2287_0070	3531	3854	323	Conserved hypothetical protein	>50%	>50%	Conserved in *Rhodococcus* spp.
IME2287_0080	3814	4413	599	Conserved hypothetical protein	>50%	>50%	Conserved in *Rhodococcus* spp.
IME2287_0090	4528	5208	680	Hypothetical protein	no match	no match	No match
IME2287_0100	5167	5409	242	Conserved hypothetical protein	>70%	>50%	Conserved in *Rhodococcus* spp.
IME2287_0110	5503	6471	968	TIR domain-containing protein	>90%	>40%	Conserved bacterial protein
IME2287_0120	6543	6731	188	Transmembrane protein	no match	no match	No match
IME2287_0130	7010	7903	893	Sigma70	>70%	>35%	Conserved in *Actinobacteria*
IME2287_0140	7900	8241	341	TPR-like domian-containing protein	no match	no match	No match
IME2287_0150	8392	9024	632	PIN domain-containing protein	>85%	>40%	Conserved in *Actinobacteria*
IME2287_0160	9073	9543	470	MerR-like DNA binding protein	>60%	60%	Conserved in *Actinobacteria*
IME2287_0170	10,014	11,267	1253	Serine recombinase	>40%	>70%	Conserved in *Rhodococcus* spp.
IME2287_0180	11,264	12,223	959	Conserved hypothetical protein	>85%	>25%	Conserved in *Actinobacteria*
IME2287_0190	12,470	12,886	416	Conserved hypothetical protein	>80%	>50%	Conserved in *Actinobacteria*
IME2287_0200	13,038	13,931	893	Conserved hypothetical protein	>90%	>40%	Conserved in *Actinobacteria*
inverted_repeat	14,284	14,313	29	-	-	-	-
direct_repeat	14,314	14,317	3	-	-	-	-

See Figure 1 for the genetic structure of the element.

**Table 3 antibiotics-12-01631-t003:** Association between genetic background, susceptibility, and the presence of pRErm46/tnErm46 and IME2287.

Source	Genetic Background	pRErm46/tnRErm46 and IME2287	pRErm46/ tnRErm46 Only	No pRErm46/tnRErm46 or IME2287	Total
Clinical	Clone 2287	40	0	0	40
	Clone G2016	1	1	0	2
	Clone G2017	0	0	0	0
	Singletons	2	6	0	8
	Susceptible	0	0	22	22
Environmental	Clone 2287	45	0	0	45
	Clone G2016	0	1	0	1
	Clone G2017	3	4	32	39
	Singletons	7	2	3	12
	Susceptible	0	0	38	38
	Total	98	14	95	207

Numbers correspond to the number of isolates found in each clonal population and carrying each genetic element combination.

**Table 4 antibiotics-12-01631-t004:** Maintenance of resistant phenotype to macrolides or rifampin in soil and during subculturing in DHS and BHI. The table exclusively presents conditions with a decrease in antimicrobial resistance phenotype.

CFU with Macrolide-Resistant Phenotype				
	DHS		Soil	
Passage	RT	37 °C	RT	37 °C
0	100%	100%	100%	100%
15	100%	99%	100%	100%
30	99%	97%	100%	100%
45	100%	64%	100%	95%

Numbers reflect the proportion of colony forming units (CFUs) that presented a macrolide-resistant phenotype after a particular number of passages under different temperature conditions (room temperature (RT) or 37 °C) and in different media (commercial donor horse serum (DHS) or autoclaved soil (Soil)). In bold are the conditions where we have seen a decrease in the resistance phenotype.

**Table 5 antibiotics-12-01631-t005:** pRErm46 and IME2287 stability in *R. equi* 2287 genetic background.

Media	Temp.	Passage	Loss of pRErm46 Only	Loss of pRErm46 and IME2287
Soil	37 °C	45	5	0
DHS	37 °C	30	1	0
	RT	15	1	0
		30	3	0
		45	13	23
Total			23	23

In the plasmid loss experiment, 46 colonies lost the macrolide resistance phenotype. The numbers in the table reflect the number of macrolide-susceptible colonies that lost pRErm46 only or pRErm46 and IME2287 under different temperature conditions (room temperature (RT) or 37 °C) and in different media (brain heart infusion (BHI), commercial donor horse serum (DHS) or autoclaved soil (Soil)). The passage in which each colony was retrieved is also indicated in the table. PCR was used to determine the presence/absence of pRErm46 and IME2287 (see materials and methods).

## Data Availability

All genomic data used in this study have been previously used in other studies and are available in the GenBank database (https://www.ncbi.nlm.nih.gov/genbank/).

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
