# Peer review of "Exploring the Accessory Genome of Multidrug-Resistant Rhodococcus equi Clone 2287"

_antibiotics, 2023, doi:10.3390/antibiotics12111631_

Round 1

Reviewer 1 Report

Comments and Suggestions for Authors

Abstract

Line 16: The molecular bases that make this MDR clone superior to the other  > The genetic mechanisms that make this MDR clone superior to the other. Do the same through the entire manuscript including line 71

Line 17: Here, we perform a deep genetic characterization of the > Here, we performed a deep genetic characterization of the

2.1. Bioinformatic analysis.

BLASTn (38) was used to align the contigs of 207 R. equi isolates characterized in our earlier studies (33, 34, 36) to known R. equi genetic elements (R. equi chromosome, pVAPA, pRErm46 and erm(51)) at >95% identity and >80% coverage. Why not also >95% for coverage?

Similarly, BLASTn (38) was used to align the unknown contigs from sample 156 to the other novel contigs and the NCBI database.  Which database exactly?

2.2. Bacterial strains and culture conditions.

Line 82: Authors need to include the name of the laboratory where these microbial culture experiments were carried out.

Line 83: Two R. equi strains were used in this study: R. equi PAM2287 (NCBI BioSample no. SAMN04880532). Which database exactly?

Line 123: PCR. Zxc , what does this mean?

Line 306: in different medias >> in different media

Line 338: We performed a deep genetic characterization of IME2287, looking for genes that could explain its persistence over time, and we found that IME2287 carries a potential 339

new virulence factor. tcpR gene encodes a putative TIR domain protein that we nominated as TcpR (Figure 1, Table 2). Authors should rewrite and punctuate this sentence for clarity. Also, consider using the word designated and noted nominated and do this throughout the entire write up.

5. Conclusions

The work presented in this manuscript studied a clonal competitive event in the zoonotic human pathogen R. equi. Consider rewriting this sentence and removing the word “manuscript”

Line 447: Patents> should be on the next separate line

Comments on the Quality of English Language

None

Author Response

Reviewer 1

We thank the reviewer for taking the time to review our manuscript and for the encouraging comments provided.

Abstract

Line 16: The molecular bases that make this MDR clone superior to the other  > The genetic mechanisms that make this MDR clone superior to the other. Do the same through the entire manuscript including line 71

ANSWER: We thank the reviewer for this comment. We rephrased the sentences following the reviewer’s advice (lines 16, 71 and 444).

Line 17: Here, we perform a deep genetic characterization of the > Here, we performed a deep genetic characterization of the

ANSWER: We thank the reviewer for this comment. We corrected the typo.

2.1. Bioinformatic analysis.

BLASTn (38) was used to align the contigs of 207 R. equi isolates characterized in our earlier studies (33, 34, 36) to known R. equi genetic elements (R. equi chromosome, pVAPA, pRErm46 and erm(51)) at >95% identity and >80% coverage. Why not also >95% for coverage?

ANSWER: We thank the reviewer for this comment. We chose a gene coverage of >80% in case certain genes in IME2287 appeared truncated.

Similarly, BLASTn (38) was used to align the unknown contigs from sample 156 to the other novel contigs and the NCBI database.  Which database exactly?

ANSWER: We used the nucleotide collection database. This information was added to the manuscript (line 79).

2.2. Bacterial strains and culture conditions.

Line 82: Authors need to include the name of the laboratory where these microbial culture experiments were carried out.

ANSWER: We thank the reviewer for this comment. This information was added to the manuscript (line 92: “All in vitro bacterial work (including the bacterial conjugation and the plasmid loss as-says) were carried out in our laboratory at the Athens Veterinary Diagnostic Laboratory from the University of Georgia (Athens, GA, USA)”).

Line 83: Two R. equi strains were used in this study: R. equi PAM2287 (NCBI BioSample no. SAMN04880532). Which database exactly?

ANSWER: We thank the reviewer for this comment. The database is indeed the BioSample database. We have specified that in the manuscript (line 83).

Line 123: PCR. Zxc , what does this mean?

ANSWER: We apologize to the reviewer. This was a typo that has been removed.

Line 306: in different medias >> in different media

ANSWER: We thank the reviewer for this comment. We corrected the typo.

Line 338: We performed a deep genetic characterization of IME2287, looking for genes that could explain its persistence over time, and we found that IME2287 carries a potential new virulence factor. tcpR gene encodes a putative TIR domain protein that we nominated as TcpR (Figure 1, Table 2). Authors should rewrite and punctuate this sentence for clarity. Also, consider using the word designated and noted nominated and do this throughout the entire write up.

ANSWER: We thank the reviewer for this comment. We rephrased the sentences following the reviewer’s advice (lines 152 and 339).

  1. Conclusions

The work presented in this manuscript studied a clonal competitive event in the zoonotic human pathogen R. equi. Consider rewriting this sentence and removing the word “manuscript”

ANSWER: We thank the reviewer for this comment. We rephrased the sentences following the reviewer’s advice (line437-439).

447: Patents> should be on the next separate line

ANSWER: We thank the reviewer for this comment. We corrected the typo.

Reviewer 2 Report

Comments and Suggestions for Authors

Using bioinformatic analysis with molecular characterization, the authors describe a genetic element in Rhodococcus equi, which possess  orthologues genes associated with antibiotic resistance/tolerance, virulence, and pathogenicity islands, bacterial persistence, and pathogen trafficking. In general, I like the work that you did, and I appreciate your effort to understand and describe IME2287, a novel genetic element in the genome of  Rhodococcus equi clone 2287. This finding is important and provides important results about the knowledge and futures perspectives of emergence and spread of zoonotic MDR-Rhodococcus equi.  Almost all collected data analysis were performed correctly.  I recommend for acceptance.

Author Response

We thank the reviewer for taking the time to review our manuscript and for the encouraging comments provided.

Reviewer 3 Report

Comments and Suggestions for Authors

The manuscript entitled "Exploring the accessory genome of multidrug-resistant Rhodococcus equi clone 2287" is well written. The study has great importance in the field of antimicrobial research. The manuscript requires minor corrections. Table 1 should be placed in the materials and method section. 

Line no. 282: Please make all tables self-explanatory. Add the full meaning of CFU when using it for the first time and check the whole manuscript for similar types of errors. 

Line no. 13: Please keep multidrug-resistant in one format either multidrug-resistant or multi-drug-resistant. Multidrug-resistant is the most common form. 

Line no. 441-445: Please make R. equi italic and check the whole manuscript for similar errors. 

Comments on the Quality of English Language

Minor editing of English language is required.

Author Response

Reviewer 3

The manuscript entitled "Exploring the accessory genome of multidrug-resistant Rhodococcus equi clone 2287" is well written. The study has great importance in the field of antimicrobial research. The manuscript requires minor corrections.

ANSWER: We thank the reviewer for taking the time to review our manuscript and for the encouraging comments provided.

Table 1 should be placed in the materials and method section. 

ANSWER: We apologize for the confusion, but Table 1 is indeed cited in materials and methods (line 130). The table was too big to be included at the end of materials and methods section, right after the paragraph where it was cited. To avoid splitting the table, we included it at the top of the next page.

Line no. 282: Please make all tables self-explanatory. Add the full meaning of CFU when using it for the first time and check the whole manuscript for similar types of errors. 

ANSWER: We thank the reviewer for this comment. We have added the full meaning of CFU in the table’s footnote for clarity.

Line no. 13: Please keep multidrug-resistant in one format either multidrug-resistant or multi-drug-resistant. Multidrug-resistant is the most common form. 

ANSWER: We thank the reviewer for this comment. We applied “multidrug-resistant” across the manuscript (lines 13, 28 and 53).

Line no. 441-445: Please make R. equi italic and check the whole manuscript for similar errors. 

ANSWER: We thank the reviewer for this comment. We verified that R. equi appeared in italics across the manuscript.